# New Material Exploration to Enhance Neutron Intensity below Cold Neutrons: Nanosized Graphene Flower Aggregation

**DOI:** 10.3390/nano13010076

**Published:** 2022-12-23

**Authors:** Makoto Teshigawara, Yujiro Ikeda, Mingfei Yan, Kazuo Muramatsu, Koichi Sutani, Masafumi Fukuzumi, Yohei Noda, Satoshi Koizumi, Koichi Saruta, Yoshie Otake

**Affiliations:** 1J-PARC Center, Japan Atomic Energy Agency, 2-4 Shirakata, Tokai-mura, Naka 319-1195, Ibaraki, Japan; 2RIKEN Center for Advanced Photonics, RIKEN, Wako 351-0198, Saitama, Japan; 3INCUBATION ALLIANCE INC. 1-2-25, Wadayama-dori, Hyogo-ku, Kobe 652-0884, Hyogo, Japan; 4Department of Materials and Analysis, Hyogo Prefectural Institute of Technology, Yukihira-cho, Suma-ku, Kobe 654-0037, Hyogo, Japan; 5Institute of Quantum Beam Science, Ibaraki University, 162-1 Shirakata, Tokai-mura, Naka 319-1106, Ibaraki, Japan

**Keywords:** nanosized material, graphene aggregation, neutron reflector, below cold neutrons, neutron source

## Abstract

It is proposed that nanosized graphene aggregation could facilitate coherent neutron scattering under particle size conditions similar to nanodiamonds to enhance neutron intensity below cold neutrons. Using the RIKEN accelerator-driven compact neutron source and iMATERIA at J-PARC, we performed neutron measurement experiments, total neutron cross-section and small-angle neutron scattering on nanosized graphene aggregation. For the first time, the measured data revealed that nanosized graphene aggregation increased the total neutron cross-sections and small-angle scattering in the cold neutron energy region. This is most likely due to coherent scattering, resulting in higher neutron intensities, similar to nanodiamonds.

## 1. Introduction

Slow neutrons, such as cold neutrons, are excellent for observing light elements such as hydrogen. They are also important nondestructive probes not only for basic physics but also for the structural genomics advancements in the life sciences and the battery technology advancements needed for the transition to a hydrogen society. Neutron-based science, also known as high-intensity-dependent science, demands that we increase the source’s intensity as high as possible. Cold neutrons are generally generated by cooling neutrons produced by nuclear reactions, spallation reactions based on accelerators, and fission reactions in nuclear reactors to a thermal equilibrium state with a hydrogenous material such as solid methane.

However, it is not easy to obtain slow neutrons below thermal equilibrium using this method, such as ultra-cold neutrons or very-cold neutrons. A new unique method focusing on nanosized particle aggregation has been proposed to increase neutron intensity in that energy region [1,2,3,4,5]. The method is based on intensity enhancement by multiple coherent scatterings of neutrons with nanosized particles. The aggregation of nanosized particles matches the wavelength of below cold neutrons, resulting in a similar effect to coherent scattering, or so-called Bragg scattering, leading to higher neutron intensities. Nanodiamonds [6,7,8,9,10] and magnesium hydride (MgH_2_) [11] have recently been reported numerically and experimentally, with the potential to increase neutron intensity by several orders of magnitude. The major challenge with nanodiamonds in practical applications is the molding method.

We focused on another carbon structure, graphene, to find a solution to this problem. Graphene [12,13,14] is expected to be the lightest and strongest material in the world due to its geometric structure. It is being scaled up for practical applications such as aircraft materials, as well as for studies on the synthesis and applications of graphene-based nanomaterials [15].

Conversely, in its development progress, large amounts of nanosized end materials are also being produced. 

We hypothesized that nanosized graphene could aid coherent neutron scattering under particle size conditions similar to nanodiamonds. Moreover, it might be possible to use it in high neutron radiation conditions due to graphene’s strong sp2 bonds. Using the RIKEN accelerator-driven compact neutron source (RANS) [16] and iMATERIA at J-PARC [17,18], we performed neutron measurement experiments such as total neutron cross-section and small-angle neutron scattering on nanosized graphene using the same method as we used for nanodiamond cross-section measurements [8].

It is also difficult to determine how much useful information can be obtained from a 700 W RANS small power source, such as total cross-section and small-angle scattering measurements, when the source intensity is overwhelmingly disadvantageous compared to MW class high-power sources like those at J-PARC [19]. As a result, the moderator arrangement system was reviewed and experimentally improved with new ideas, such as selecting a slab-type moderator arrangement and incorporating slit-type collimators to increase available neutrons [20], as one way of finding out the potential of small sources [21].

In this paper, we report the potential of nanosized graphene as a reflector material below cold neutrons, together with experimental results.

## 2. Nanosized Graphene Flower Aggregation

A combination of phenolic formaldehyde resin carbonization and the hot isostatic pressing (HIP) process produces a distinctive graphene called a “graphene flower” with a “sunflower” shape [22,23]. Note that this graphene flower is far away from how we imagine graphene as a two-dimensional material. The produced graphene flower, as shown in Figure 1, was composed of multiple petal-like graphene layers of less than 20 nm thick growing from the carbonized sphere surface. Seed-like graphene was composed of many 1–100 nm sized graphenes inside the sphere, forming a concave–convex structure. Finally, graphene flowers combined to form a nanosized graphene aggregate.

For this study, two graphene sample types, namely, graphene 1 and 2, were prepared under different HIP conditions. First, a phenol formaldehyde resin, BellPearl S870, manufactured by Air Water Co., Ltd., with an average particle size of 15 μm, was prebaked in a constant flow of nitrogen gas at a maximum temperature of 650 °C. The heat-treated powder was placed in a graphite crucible and subjected to HIP in argon gas at a maximum ultimate pressure of 70 MPa. Graphene 1 was heated to a maximum temperature of 1390 °C at a rate of 900 °C/h, held there for 1 h, and then cooled. Graphene 2 was heated to a maximum temperature of 1300 °C at a rate of 200 °C/h and then cooled immediately without any retention time.

As shown in Figure 2a, graphene 1 produced petal-like graphene on the surface and granular seed-like graphene inside the sphere. Graphene 2 formed petal-like graphene but almost no seed-like graphene, as shown in Figure 2b. The samples were shaped to 20 mm in diameter × 2 mm in thickness. The samples were irradiated with neutrons after being wrapped in a single layer of commercially available aluminum foil (12 μm thick). For comparison, the nanodiamond used in the literature [8] and the graphite were similarly irradiated. Table 1 summarizes the samples used in the experiments.

## 3. Experiment

### 3.1. Neutron Transmission

The experiment was carried out using the RANS neutron source, which was generated by the p-Be reaction of a 7 MeV proton bombardment, as shown in Figure 3. The neutron transmission experiment’s basic experimental methods and data processing were based on the literature [8]. Neutron transmission experiments generally use parallel neutron beams to determine the total neutron cross-section. However, in the presence of a low-intensity source of RANS and to improve statistical accuracy in a short-time measurement at the expense of spatial resolution, the “slit-type collimator” concept was developed, in which the neutron beam emitted from the moderator surface was gradually focused by placing boron carbide (B_4_C) slits in the locations shown in Figure 3. The beam was gradually narrowed down to a final diameter of 10 mm using this collimator. The sample was placed at the exit of the final slit, corresponding to a position of approximately 1.4 m from the mesitylene moderator emission surface.

Neutron transmission was measured using the two-dimensional neutron detector with Gas Electron Multiplier, called an n-GEM [24] detector, which was installed approximately 80 mm from the sample. The resulting neutron intensity was approximately two orders of magnitude higher than that in the parallel beam case. The neutron spectral intensities at the detector positions of the J-PARC BL-10 [8] and RANS through slit-type collimators are shown in Figure 4 for comparison.

Although this is only a relative comparison due to the different moderator and detector positions, the mesitylene moderator’s cold neutron component, forming a peak at 3–5 meV, was confirmed.

### 3.2. Small-Angle Neutron Scattering (SANS) Experiment

SANS experiments were performed at the J-PARC iMATERIA [17,18]. Measurements were also performed on a small-angle scattering instrument [25] under development at RANS as a comparison.

## 4. Results and Discussions

### 4.1. Neutron Transmission

Figure 5 depicts the transmission measurement results. Each measured data was normalized per atom from sample volume, density, and composition. Compared to graphite, nanodiamond had a lower transmission below the meV region. Graphene 1 showed low transmittance, close to that of the nanodiamond. Graphene 2 had somewhat higher transmission than graphene 1. This may be due to different sample fabrication conditions. However, graphite Bragg cutoff peaks at approximately 1.6 meV were also measured in graphenes 1 and 2. This is because during the HIP treatment of the carbonized phenolic resin to form graphene, the formation of the graphite surface into nanosized graphene was not always sufficient, and the graphite components remaining in the material were detected, leaving room to find the optimum nanosized graphene formation conditions. Total neutron cross-sections were calculated using the transmission data, as shown in Figure 6, where coherent scattering predominated for the 0.5–2 meV region, concerning the nanodiamond literature [8].

Compared to graphite, nanodiamond had several hundred higher total neutron cross-sections below the meV region. Graphene 1 was approximately one-third of the total neutron cross-section data of nanodiamonds. Conversely, it was several tens of times higher than graphite. The findings indicate that nanosized graphene, which is still in the early development stages, has the potential to rival or even surpass nanodiamonds with further fabrication process optimization, including nanosized graphene aggregation.

### 4.2. SANS

#### 4.2.1. SANS Results

Figure 7 shows SANS, measured on the diffractometer iMATERIA at J-PARC [17,18]. We recognized Bragg’s scattering peaks in the q range of 1–10 Å^−1^ for SANS obtained for graphenes 1 and 2, originating from interlayer interference. The SANS profiles obtained for graphenes 1 and 2 showed broad shoulders at q = 0.15 Å^−1^, attributed to a graphite primary particle sheet [26]. The shoulder q-positions were translated to 4.8 nm (=2π/q). For nanodiamonds, the primary particle sheet size was evaluated at approximately 10 nm. According to Porod’s law [26], the tail of the shoulders decays according to q^−4^, indicating a smooth surface. Compared to nanodiamonds, and graphenes 1 and 2, the SANS obtained for graphite showed a strong upturn obeying q^−4^ toward lower q = 0.01 Å^−1^, indicating a larger particle sheet size. It should be denoted that for graphenes 1 and 2, we could not recognize the fringes or oscillations in the q-behavior. This is due to the large inhomogeneity in shape and size of the graphenes.

The SANS profile of graphene 1 was also measured using the RANS-developed small-angle scattering instrument (ib-SAS [25]) (see open circles in Figure 7). Even though the SANS for graphene 1 obtained at RANS had a narrow q-region ranging from 0.02 to 0.2 Å^−1^, the q-behavior was consistent with that obtained at the iMATERIA. The ib-SAS instrument is still being developed to reduce the noise background.

#### 4.2.2. Invariant Q for SANS Data

Invariant *Q*, representing the scattering cross-section due to small-angle scattering [27], was evaluated as follows,
(1)$$Q=\frac{1}{2\pi^{2}}\int \;q^{2}I\left(q\right)\;dq$$
by using iMATERIA and the q-region of 0.007–0.45 Å^−1^. Table 2 shows the *Q* values. In particular, nanodiamond and graphene 1 showed a larger *Q*. These findings agree with the scattering cross-section values determined by transmission experiments.

## 5. Conclusions

We concentrated on nanosized graphene aggregation to intensify below cold neutrons, which are similar to those of nanodiamonds. Using the RANS, we performed neutron measurement experiments, total neutron cross-section, and SANS on nanosized graphene aggregation. For the first time, the measured data revealed that nanosized graphene aggregation increased total neutron cross-sections and small-angle scattering in the cold neutron energy region, most likely due to coherent scattering, implying the possibility of below cold neutron applications. We will go on to focus on the radiation resistance of graphene, which is expected from the strong sp2 bond.

## Figures and Tables

**Figure 1 nanomaterials-13-00076-f001:**
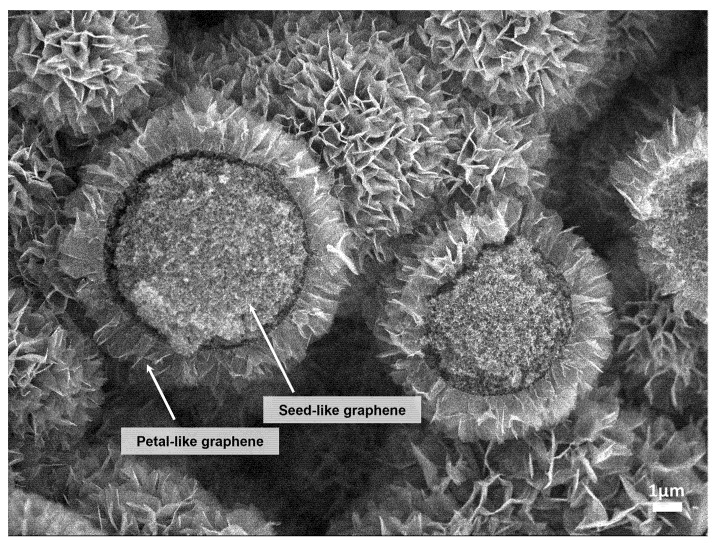
The SEM image of a graphene flower produced by the HIP treatment. As indicated by the arrows in the figure, the graphene flower is composed of multiple petal-like graphene and seed-like graphene. Petal-like graphene forms multiple layers of petals less than 20 nm thick growing from the carbonized sphere surface. The seed-like graphene is composed of many 1–100 nm sized graphenes inside the sphere, forming a concave–convex structure.

**Figure 2 nanomaterials-13-00076-f002:**
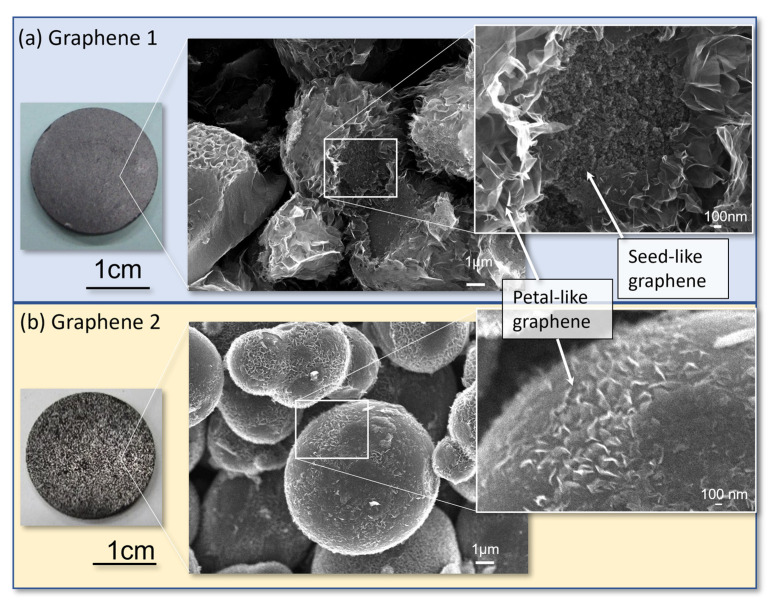
Photographs of graphene materials prepared under various HIP conditions. (**a**) Graphene 1, which formed petal-like graphene and nm-sized granular seeds, called “seed-like graphene”, was fabricated under HIP conditions with a maximum temperature of 1390 °C at a rate of 900 °C/h, 1 h holding time, then cooled naturally; (**b**) Graphene 2, which formed petal-like graphene, but no seed-like graphene, was fabricated under HIP conditions with a maximum temperature of 1300 °C at a rate of 200 °C/h and then cooled immediately without any retention time.

**Figure 3 nanomaterials-13-00076-f003:**
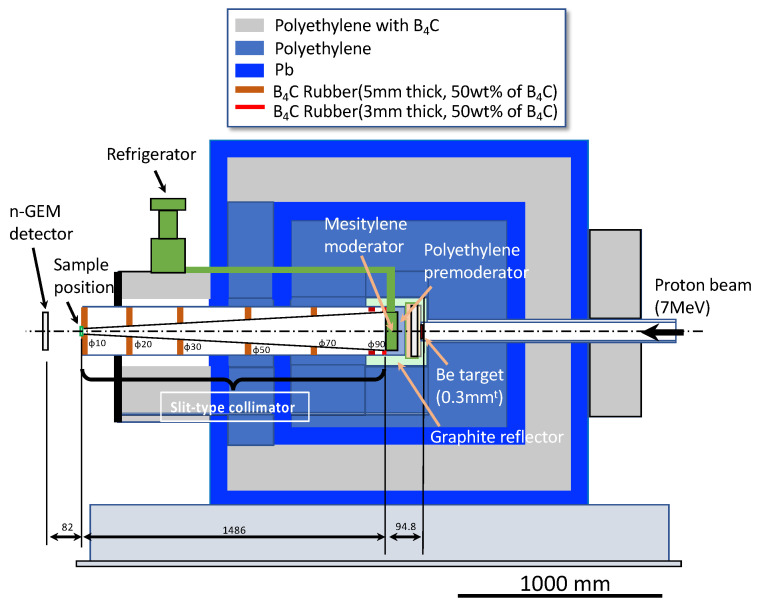
RANS target system cross-section. In a slab-type mesitylene moderator arrangement neutron beamline, a slit-type collimator was installed to increase available neutron intensity at the sample position.

**Figure 4 nanomaterials-13-00076-f004:**
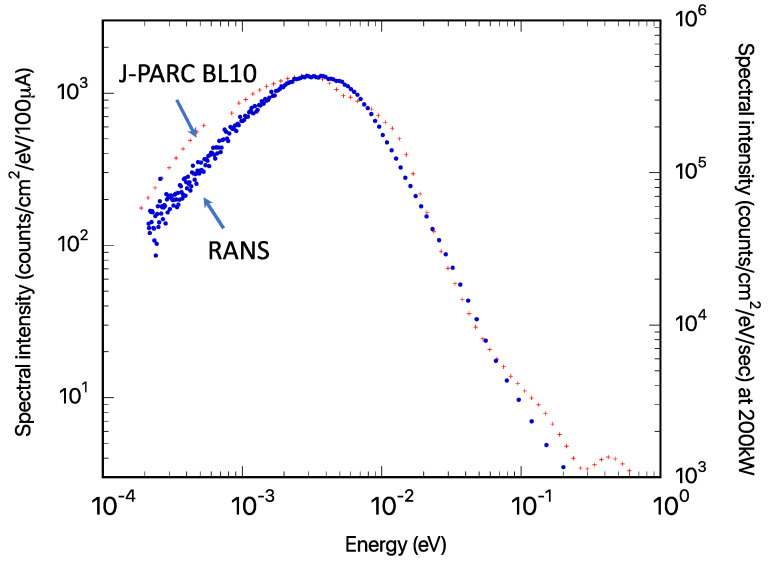
Neutron spectral intensity from slab-type mesitylene moderator at the sample position at RANS. The sample is 1.49 m away from the moderator. The neutron spectral intensity from decoupled liquid hydrogen moderator at the J-PARC BL10 at the detector position, which was approximately 14.7 m from the moderator, is also shown for reference.

**Figure 5 nanomaterials-13-00076-f005:**
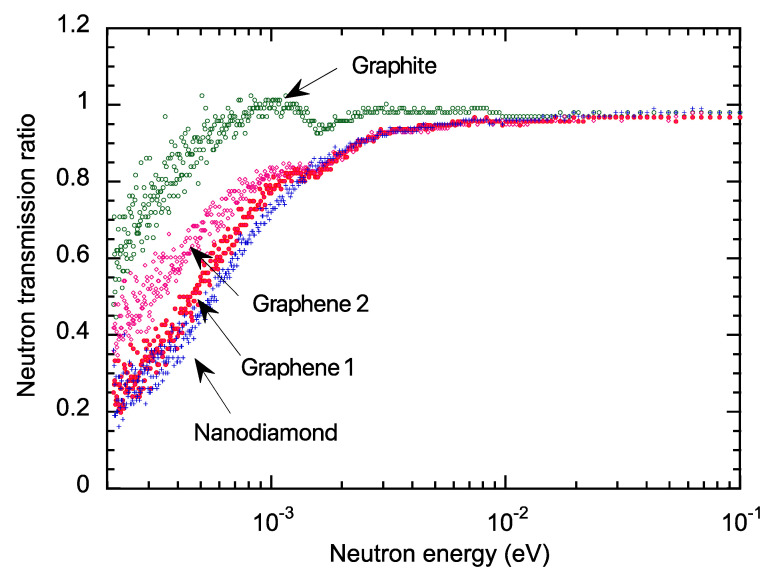
Energy-dependent neutron transmission measurements. Compared to graphite, nanodiamond had lower transmission below the meV region. Graphene 1 showed low transmittance, close to that of nanodiamond. Graphene 2 had somewhat higher transmission than graphene 1.

**Figure 6 nanomaterials-13-00076-f006:**
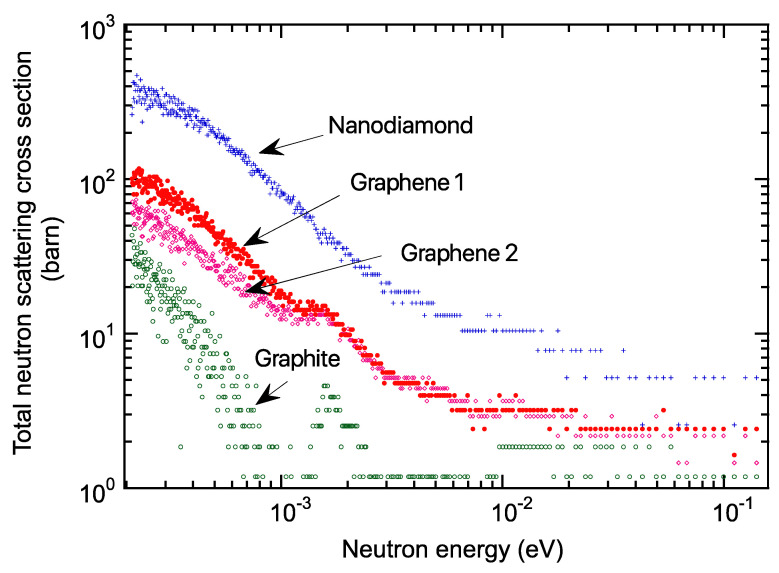
Total neutron cross-section. In comparison with graphite, nanodiamond had several hundred higher total neutron cross-sections below meV region. Graphene 1 was approximately one-third of the total neutron cross-section data of nanodiamond. Conversely, it was several tens of times higher than graphite.

**Figure 7 nanomaterials-13-00076-f007:**
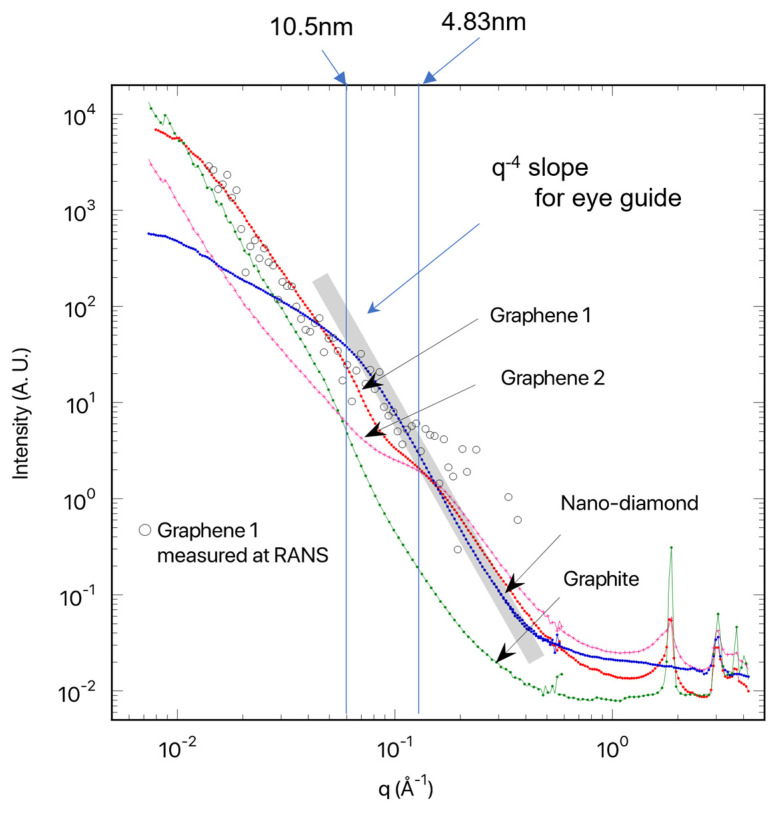
Small-angle neutron scattering measurements using iMATERIA at J-PARC. Bragg’s scattering peaks were recognized in the q range of 1–10 Å^−1^ for graphenes 1 and 2, originating from interlayer interference. Compared to nanodiamond, and graphenes 1 and 2, the SANS obtained for graphite shows a strong upturn obeying q^−4^ toward lower q = 0.01 Å^−1^, indicating a larger particle sheet size. It should be denoted that for graphenes 1 and 2, we cannot recognize the fringes or oscillations in the q-behavior, indicating the large inhomogeneity in shape and size of the graphenes. Open circles in the figure show measured data using the RANS-developed small-angle scattering instrument, ib-SAS.

**Table 1 nanomaterials-13-00076-t001:** Shape, thickness, and density of graphene, graphite, and nanodiamond materials.

Sample	Graphene 1	Graphene 2	Graphite	Nanodiamond
Shape	ϕ20 mm Disk			10 × 10 mm Plate
Thickness (mm)	2.6	3.2	1.9	1.2
Bulk density (g/cm^3^)	1.09	0.91	1.77	0.65

**Table 2 nanomaterials-13-00076-t002:** Calculated invariant Q.

Sample	Graphene 1	Graphene 2	Graphite	Nanodiamond
Invariant Q	0.0213	0.0114	0.0081	0.0164

## Data Availability

The data presented in this study are available on request from the corresponding authors.

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
