# Peer review of "New Material Exploration to Enhance Neutron Intensity below Cold Neutrons: Nanosized Graphene Flower Aggregation"

_nanomaterials, 2022, doi:10.3390/nano13010076_

Round 1

Reviewer 1 Report

The manuscript introduces graphene an alternative material for enhancing albedo neutron scattering enhanced by small-angle scattering effects. 

Total cross section measurements were conducted at RANS with an unconventional arrangement of source, sample and detector, which rises the question of the quantification of scattering contributions picked up by the detector. For graphite one expects a total cross section of around 5 barns at the higher energies, which is not what came out in the measurement. It is somewhat surprising that the total cross section of all the measured carbon-based materials are not settling on the same value at 0.1 eV.

Reviewer 2 Report

This is an interesting article based on an original idea. The research design is clearly explained. The use of RANS for the samples characterization is of interest. I recommend to publish this article after minor revision. 

particular remarks:

- The authors state the good resistivity of graphene to high radiation fluxes. This statement has to be proved experimentally. Please, give the result of such a study, or explain how you are going to perform such a test in the future. Otherwise, avoid giving such strong statements without proving them. 

- Evidently, the performance of a graphene sample for neutron scattering depends on the conditions of its preparation thus on its parameters. They do not seem to be under sufficient control in the current study. Please, explain your research plan needed to improve the sample characterization/preparation.

- Figure captions have to be self-sufficient. Please, give more detail.
